

# The complete chloroplast genome of *Fagus crenata* (subgenus *Fagus*) and comparison with *F. engleriana* (subgenus *Engleriana*)

James R. P. Worth[1], Luxian Liu[2], Fu-Jin Wei[1] and Nobuhiro Tomaru[3]

[1] Department of Forest Molecular Genetics and Biotechnology, Forestry and Forest Products Research Institute, Tsukuba, Japan
[2] Key Laboratory of Plant Stress Biology, Laboratory of Plant Germplasm and Genetic Engineering, College of Life Sciences, Henan University, Kaifeng, China
[3] Graduate School of Bioagricultural Sciences, Nagoya University, Nagoya, Japan

## ABSTRACT

This study reports the whole chloroplast genome of *Fagus crenata* (subgenus *Fagus*), a foundation tree species of Japanese temperate forests. The genome has a total of 158,227 bp containing 111 genes, including 76 protein-coding genes, 31 tRNA genes and 4 ribosomal RNA genes. Comparison with the only other published *Fagus* chloroplast genome, *F. engeleriana* (subgenus *Engleriana*) shows that the genomes are relatively conserved with no inversions or rearrangements observed while the proportion of nucleotide sites differing between the two species was equal to 0.0018. The six most variable regions were, in increasing order of variability, *psb*K-*psb*I, *trn*G-*psb*fM, *rpl*32, *trn*V, *ndh*I-*ndh* and *ndh*D-*psa*C. These highly variable chloroplast regions in addition to 160 chloroplast microsatellites identified (of which 46 were variable between the two species) will provide useful genetic resources for studies of the inter- and intra-specific genetic structure and diversity of this important northern hemisphere tree genus.

## INTRODUCTION

The genus *Fagus* is a major tree of temperate forests of the northern hemisphere with two informal subgenera recognized (*Shen, 1992*): *Engleriana* with three species and *Fagus* with seven species (*Oh, 2015*; *Renner et al., 2016*). The genus has been the focus of intensive genetic studies over the last 30 years enabling insights into relationships of the extant species (*Denk, Grimm & Hemleben, 2005*), the impact of the interglacial-glacial cycles on extant genetic diversity (*Fujii et al., 2002*; *Magri et al., 2006*) and predictions of the impacts of ongoing climate change (*Csilléry et al., 2014*). However, despite the significance of the genus there remains a dearth of Next Generation Sequencing based-genetic resources for *Fagus*, including for the chloroplast genome, with the whole chloroplast genome of only a single species, the Chinese endemic *F. engleriana* of subgenus *Engleriana* (*Yang et al., 2018*), so far published.

Corresponding author
James R. P. Worth,
jrpw2326@affrc.go.jp

This study reports the whole chloroplast genome of the Japanese endemic *Fagus crenata*, the first reported of subgenus *Fagus*. This species is a foundation tree of Japan's cool temperate forest ecosystem and is distributed widely from the mountains of southern Kyushu (31.4° N 130.8° E) to southern Hokkaido (42.8° N 140.2° E). Phylogeographic studies based on Sanger sequencing of small portions of the chloroplast genome have revealed strong geographic structuring of chloroplast haplotypes (*Fujii et al., 2002*; *Okaura & Harada, 2002*) that, combined with fossil pollen data (*Tsukada, 1982*), suggests the species persisted in multiple coastal refugia and has occupied most of its current wide geographic range in the postglacial. Here we report the whole chloroplast genome sequence of *F. crenata* and compare it to the genome of *F. engleriana* (subgenus *Engleriana*). This data will be a useful genetic resource for investigating the phylogenetic relationship of *Fagus* and for developing chloroplast-based genetic markers, including both single nucleotide polymorphism- and microsatellite-based markers.

## MATERIALS AND METHODS

### Next Generation Sequencing and chloroplast genome assembly

Whole genomic DNA was extracted from a single sample of *F. crenata* collected from Daisengen Peak, Hokkaido, Japan (41.616° N–140.1333° E) representing the *F. crenata* chloroplast haplotype A (following *Fujii et al., 2002*) using a modified CTAB protocol (*Doyle, 1990*). DNA concentration and quality were assessed by agarose gel electrophoresis and a Qubit 2.0 fluorometer (Life Technologies). A total of 9 μg of DNA was sent to the Beijing Genomic Institute where short-size Truseq DNA libraries were constructed and paired-end sequencing (2 × 100 bp) was performed on an Illumina HiSeq2000 Genome Analyser resulting in a total of 7,223,910 reads (the raw sequence reads are deposited in NCBI BioProject Database Accession number: PRJNA528838).

Assembly of chloroplast DNA from the whole genomic sequencing data was undertaken in Novoplasty 2.6.3 (*Dierckxsens, Mardulyn & Smits, 2016*), a seed-and-extend algorithm that is designed for the specific purpose of assembling chloroplast genomes from whole genome sequencing data, starting from a chloroplast seed sequence (*trnK-matK* of haplotype A: Genbank accession AB046492). This resulted in nine chloroplast contigs varying in length from 2,748 to 43,982 bp constructed from 230,360 chloroplast reads (3.19% of the total reads) with an average read coverage of the chloroplast genome of 145. The nine contigs were ordered and oriented using the *F. engleriana* whole chloroplast genome (KX852398) as a reference and the complete chloroplast sequence of *F. crenata* was constructed by connecting overlapping terminal sequences. Sanger sequencing was undertaken to check the accuracy of assembly of the joins of the nine contigs and the inverted repeat and single copy regions and also the sequences of the most diverged sites between *F. crenata* and *F. engleriana* (see "Results and Discussion"). A total of 8,146 bp was sequenced using 15 primer pairs and no differences were observed with the *F. crenata* genome apart from those due to inaccurate sequence at the terminal ends of the Sanger sequences.

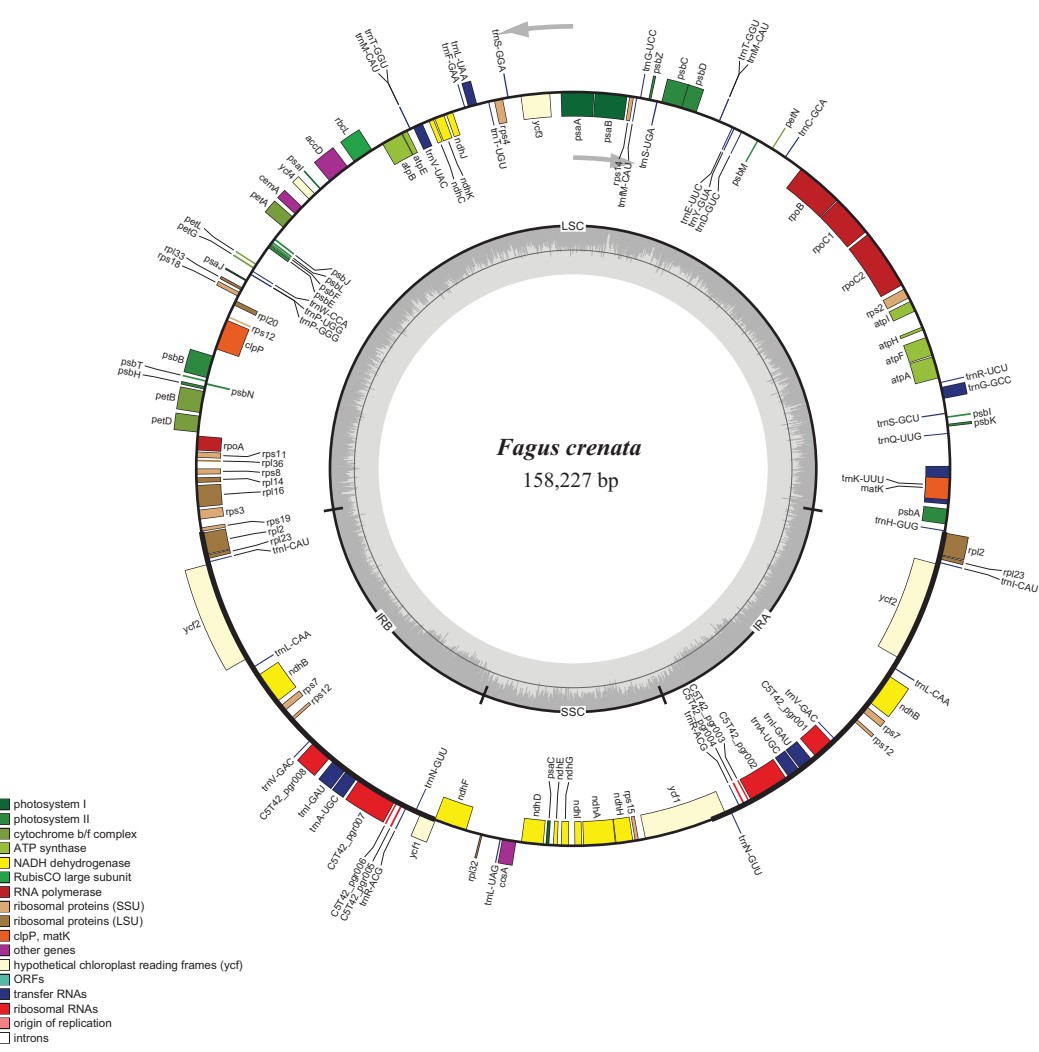

**Figure 1 Chloroplast genome map of *Fagus crenata*.** Genes inside the circle are transcribed clockwise, genes outside are transcribed counter-clockwise. The light gray inner circle corresponds to the AT content, the dark gray to the GC content. Genes belonging to different functional groups are shown in different colors. The position of the large single copy (LSC) region, small single copy (SSC) region and two inverted repeats (IRa and IRb) are also indicated.

## Chloroplast genome annotation

The annotation of the chloroplast genome was performed using the online program Dual Organellar Genome Annotator (*Wyman, Jansen & Boore, 2004*). Initial annotation, putative starts, stops and intron positions were determined according to comparisons with homologous genes of *F. engleriana* chloroplast genome using Geneious v9.0.5 (Biomatters, Auckland, New Zealand). A circular gene map was drawn by the OrganellaGenomeDRAW tool (OGDRAW) followed by manual modification (*Lohse, Drechsel & Bock, 2007*).

## Phylogenetic analysis and assessment of divergent regions

A multiple sequence alignment of *F. crenata*, *F. engleriana*, representative whole chloroplast genomes of the Fagaceae family and outgroups from Betulaceae, Juglandaceae and
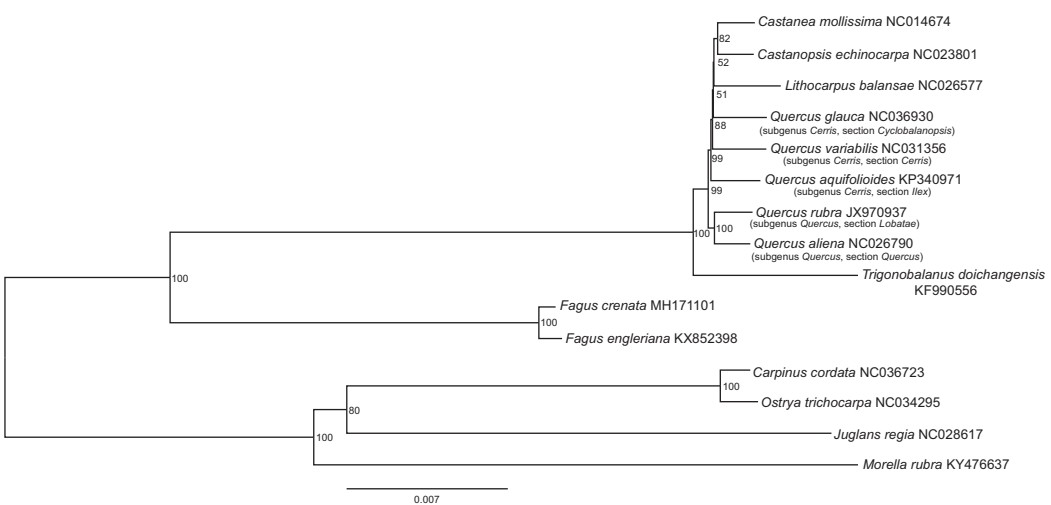

**Figure 2 The best scoring maximum likelihood phylogenetic tree based on 143,882 bp of non-gapped sequence in a Gblock alignment of the whole chloroplast genomes of *Fagus crenata*, *F. engleriana*, representative genera of the Fagaceae and outgroups from Betulaceae, Juglandaceae and Myricaceae.** A total of 11.48% of the sites were variable. The Genbank accession number of each chloroplast genome is shown after the species name.

Myricaceae obtained from Genbank was constructed using T-Coffee using default parameters (*Notredame, Higgins & Heringa, 2000*). Subsequently, Gblocks v0.91b (*Castresana, 2000*) was used to identify homologous blocks of DNA and remove poorly aligned and divergent regions of the chloroplast genomes. RAxML NG (*Kozlov et al., 2018*) was then used to construct a maximum likelihood phylogenetic tree using the most appropriate DNA substitution model, TVM+I+G, as estimated in jModelTest 2.1.10 (*Darriba et al., 2012*) and 1,000 bootstrap replicates.

Pairwise nucleotide differences (p-distance) between the sequences of the Gblocks alignment were calculated in Mega 7 (*Kumar, Stecher & Tamura, 2016*) excluding parts of the sequence alignment with gaps. The coding genes, non-coding regions and intron regions were compared between the alignment of the two *Fagus* chloroplast genomes to detect divergence hotspots. We examined 101 regions (39 coding genes, 52 intergenic spacers and 10 intron regions) of the two *Fagus* species for nucleotide variability (Pi) values calculated in DnaSP v5.0 (*Librado & Rozas, 2009*).

## Identification of chloroplast microsatellites

Chloroplast microsatellite regions shared in both *F. crenata* and *F. engleriana* were searched for in an alignment of the two full chloroplast genomes (constructed by MAFFT v7.308 (*Katoh et al., 2002*) under default settings) using Phobos Tandem Repeat Finder (*Mayer, 2008*) implemented in Geneious v9.0.5. Microsatellite in either of the sequences with a repeat unit length of 1–2 bp were searched for using a minimum length of 10 bp while those with a repeat length of 3–6 bp were selected if they displayed three or more repeats.

## RESULTS AND DISCUSSION

The assembled whole chloroplast genome of *F. crenata* has a total of 158,227 bp (Fig. 1: Genbank accession number MH171101) and consisted of an 87,557 bp large single copy

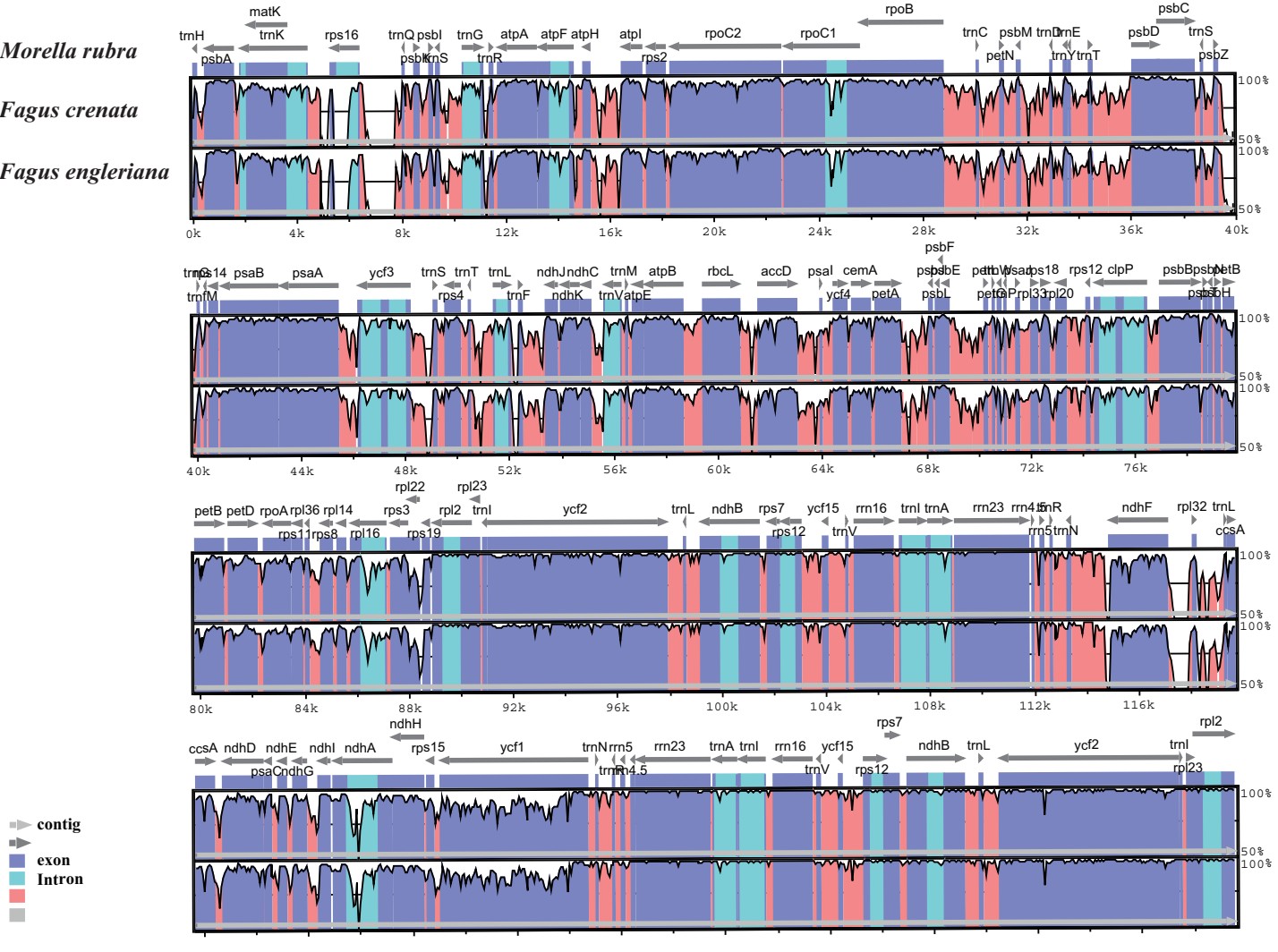

**Figure 3 Visualization of alignment of the two *Fagus* chloroplast genome sequences, with *Morella rubra* (Myricaceae, Fagales) as a reference.** The horizontal axis indicates the coordinates within the chloroplast genome. The vertical scale indicates the percentage of identity, ranging from 50 to 100%. Genome regions are color coded as protein coding, intron, mRNA and conserved non-coding sequence (CNS).

region, a 18,928 bp small single copy region and two inverted repeats 25,871 bp in length. The genome contained 111 genes, including 76 protein-coding genes, 31 tRNA genes and 4 ribosomal RNA genes (see DatasetS1 for the genbank file of the chloroplast genome). The Gblocks alignment consisted of 143,882 bp of non-gapped sequence of which 11.48% of sites were variable (see DatasetS2 for Gblocks alignment). The resulting best ML tree had similar relationships to previous studies with *Fagus* as sister to all other Fagaceae (*Manos & Steele, 1997*) (Fig. 2). *Fagus crenata* and *F. engleriana* formed a strongly diverged clade consistent with previous evidence of the large divergence of *Fagus* from all other Fagaceae genera (*Heenan & Smissen, 2013*). The proportion of nucleotide sites that differed (p-distance) between *F. crenata* and *F. engleriana* was 0.0018 which was lower than any other pairwise differences observed including between five *Quercus* species which

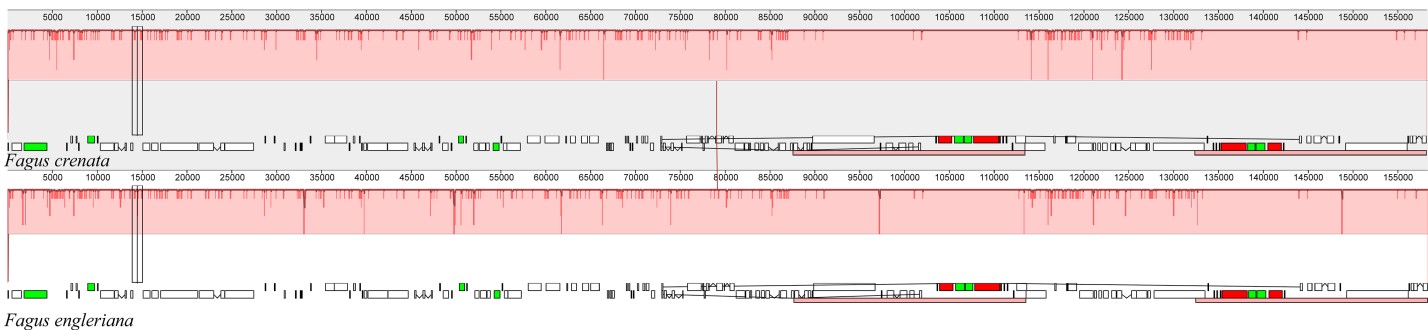

**Figure 4 A MAUVE (*Darling et al., 2004*) alignment of *Fagus crenata* and *F. engleriana* chloroplast genomes showing the lack of rearrangements between the chloroplast genomes of the two species.** The *Fagus crenata* genome is shown at top as the reference. Within each of the alignment, local collinear blocks are represented by blocks of the same color connected by lines.

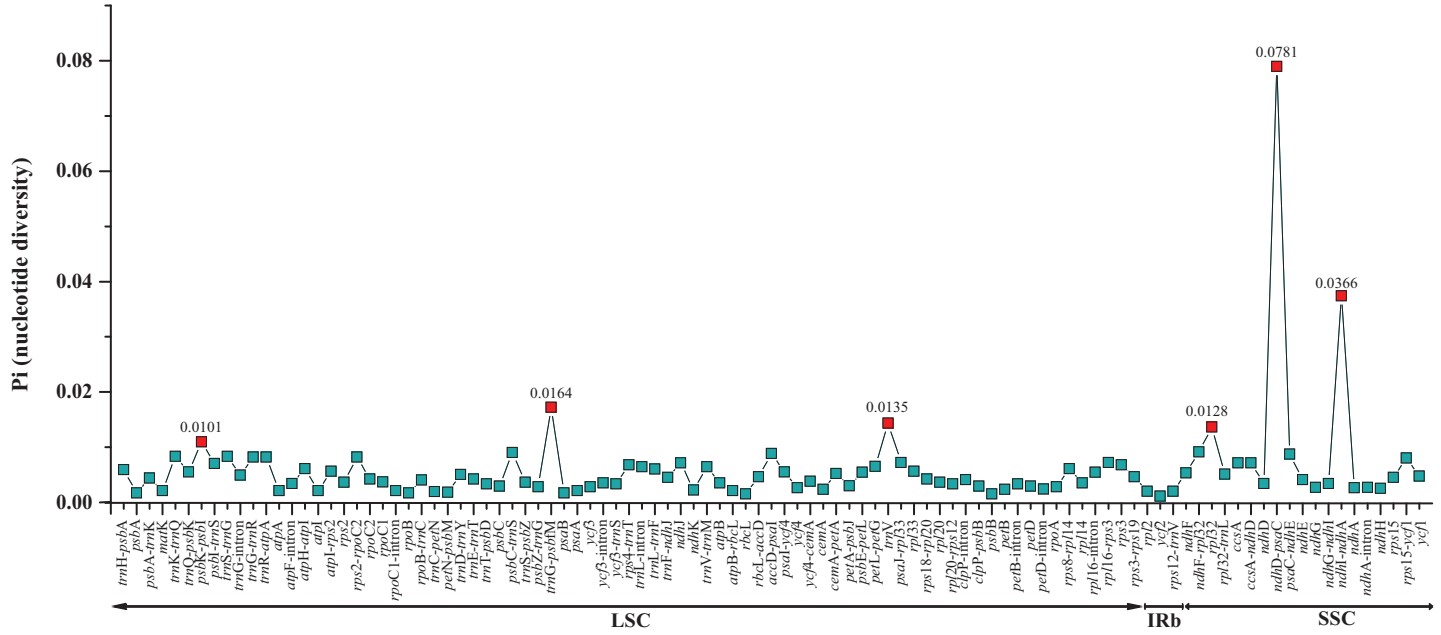

**Figure 5 Comparative analysis of the nucleotide diversity (Pi) values between the 101 regions (39 coding genes, 52 intergenic spacers, and 10 intron regions) of the whole chloroplast genomes of the *Fagus crenata* and *F. engleriana*.** The regions are ordered based on their position in the chloroplast genomes.

had values between 0.0035 and 0.0047 (average = 0.0042) (see DatasetS3 for a matrix of p-distances).

The two *Fagus* chloroplast genomes were relatively conserved (Fig. 3) with the IR region more conserved than both the large single copy (LSC) and small single copy (SSC) regions. We did not detect either inversions or translocations among the two genome sequences, and no rearrangement occurred in gene organization after verification (Fig. 4). There was high variation in nucleotide diversity values observed between the 101 regions of the two *Fagus* species with values ranging from 0.0003 (*ycf*2 gene) to 0.0781 (*ndh*D-*psa*C) (Fig. 5). The six most variable regions were, in increasing order of variability, *psb*K-*psb*I, *trn*G-*psb*fM, *rpl*32, *trn*V, *ndh*I-*ndh* and *ndh*D-*psa*C of which four are located in

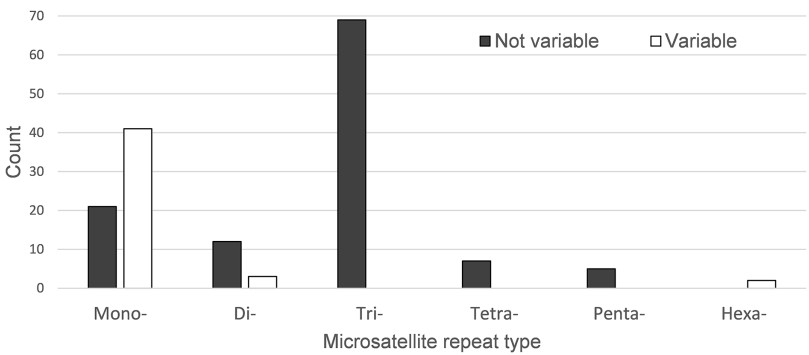

**Figure 6 The number of chloroplast microsatellites of each repeat type shared in *Fagus crenata* and *F. engleriana*.** Microsatellites that displayed no size variation between the two species are indicated by black bars while those that did are indicated by white bars. Note that the number of variable tri-, tetra- and penta-nucleotide chloroplast microsatellites was zero.

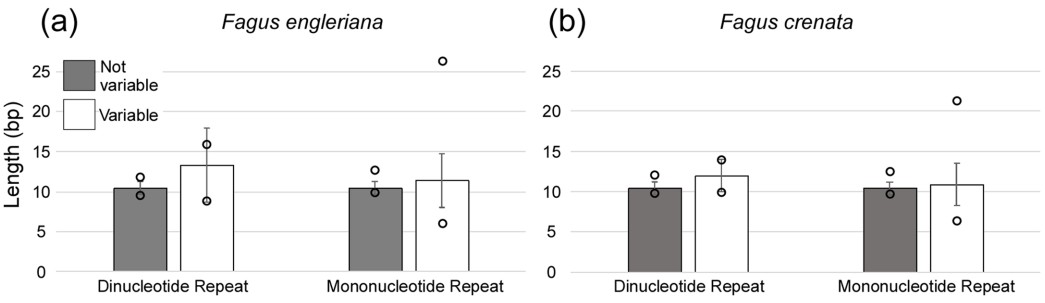

**Figure 7 The average length (bp) of variable versus non-variable chloroplast microsatellites for both mono- and di-nucleotide repeat motif types observed in both (A) *Fagus crenata* and (B) *F. engleriana* including the standard deviation (error bars) and minimum and maximum lengths (empty circles).**

the LSC region and two in the SSC region (Fig. 5). The nucleotide diversities of these variable regions between *F. crenata* and *F. engleriana* were higher than observed within some other studies of Fagaceae genera including East Asian (*Yan et al., 2018*) and Mediterranean oaks (*Vitelli et al., 2017*).

A total of 160 chloroplast microsatellites with a repeat unit length between 1 and 6 bp were identified based on the selection criteria in the two species of which mono- and tri-nucleotide repeat microsatellites were the most abundant with a frequency of 38.7% and 43.1%, respectively. This abundance of mono- and tri-repeats in the chloroplast is similar to a range of other angiosperms (*Melotto-Passarin et al., 2011*). Of these microsatellites, 46 displayed size variation between *F. crenata* and *F. engleriana* (see DatasetS4 for a table with details of all 46 variable chloroplast microsatellites). The majority (66.1%) of the variable chloroplast microsatellites were mono-nucleotide repeats while 20% of di-nucleotide repeats and both of the two hexa-nucleotide repeats were variable. On the other hand, zero of the tri-, tetra- and penta-nucleotide repeats showed size variation between the two species (Fig. 6). The length of variable versus non-variable chloroplast microsatellites was similar but with a greater length variation for variable microsatellites in both *F. crenata* and *F. engleriana* (Fig. 7).

## CONCLUSION

Overall, the chloroplast genome of *F. crenata* will provide a useful genetic resource for future genetic studies into the foundation temperate tree genus *Fagus*. Specifically, the chloroplast genomes of both informal subgenera will provide useful references and sources of molecular markers to investigate phylogeographic patterns of the chloroplast within and between *Fagus* species. Some major questions are yet to be resolved in *Fagus*, including resolving taxonomic boundaries of western Eurasian *Fagus* populations which has remained a recalcitrant problem due to low marker resolution and high within-species genetic diversity (*Denk et al., 2002*) and the non-monophyly of the chloroplast of East Asian species as suggested by Sanger sequence-based data (*Manos & Stanford, 2001*; *Okaura & Harada, 2002*).

## ACKNOWLEDGEMENTS

We would like to thank fellow lab members for their advice on this study and H. Kanehara for her assistance in the lab.

### Funding

This work was supported by the Japanese Society for the Promotion of Science Grant-in-Aid for Young Scientists A (Grant number 16748931); and a Forestry and Forest Products Research Institute grant (Grant number 201430). The funders had no role in study design, data collection and analysis, decision to publish, or preparation of the manuscript.

### Grant Disclosures

The following grant information was disclosed by the authors:
Japanese Society for the Promotion of Science Grant-in-Aid for Young Scientists A: 16748931.
Forestry and Forest Products Research Institute grant: 201430.

### Competing Interests

The authors declare that they have no competing interests.

### Author Contributions

- James R. P. Worth conceived and designed the experiments, performed the experiments, analyzed the data, prepared figures and/or tables, authored or reviewed drafts of the paper, approved the final draft.
- Luxian Liu conceived and designed the experiments, analyzed the data, prepared figures and/or tables, authored or reviewed drafts of the paper, approved the final draft.
- Fu-Jin Wei analyzed the data, prepared figures and/or tables, approved the final draft.
- Nobuhiro Tomaru conceived and designed the experiments, contributed reagents/materials/analysis tools, authored or reviewed drafts of the paper, approved the final draft.
## DNA Deposition

The following information was supplied regarding the deposition of DNA sequences:

Data are available at GenBank, accession number: MH171101.

## Data Availability

Data are available at the BioProject database:

BioProject ID: PRJNA528838.

## Supplemental Information

Supplemental information for this article can be found online at http://dx.doi.org/10.7717/peerj.7026#supplemental-information.

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
