# Peer review of "The complete chloroplast genome of Fagus crenata (subgenus Fagus) and comparison with F. engleriana (subgenus Engleriana)"

_PeerJ, doi:10.7717/peerj.7026_

## Round 0.1 · original submission · Major Revisions

Dear author

Your paper has been assessed by three reviewers and myself as academic Editor.

As you could see below, the quality of the methods and the results were rated positively by all the reviewers. Only one reviewer did not consider that the findings justified a full paper publication. But the journal phylosophy of PeerJ is not to judge on the criteria of novelty or subjective impact, but on the statistical rigor and reliability of the data. And in this regard the work presented is of excellent quality, So as academic editor I do suggest to expand and improve the discussion section.

I congratulate you for the nice piece of work, which will add value to PeerJ.

Please address the main concerns of the reviewers and submit a revised version of the manuscript. Please also include a response to each reviewer. I also urge you to deposit the original short reads from their NGS experiments on an appropriate database. The Multiple Sequence Alignments (MSA) of the full chloroplasts should be included as supplementary data.

with kind regards

Reviewer 1 ·

Basic reporting

This is well written report of a chloroplast genome. The work is new and well performed. However there are now dozens of chloroplasts genomes in databases, they are very well conserved as it is in this case. In my opinion the results should be part of a more comprehensive genome analysis if Fagus species or it should be downloaded in a database. There is no reason for a specific publication

Experimental design

This is a well performed work, well described methods and no special comments

Validity of the findings

This is a report on a chloroplast genome from a new species that just confirms that in this case is also very well conserved. They found 0.197% of differences between the two species studied. This finding is not a reason for a full paper publication

Additional comments

The result should be put in a database or to be included in a more comprehensive analysis of Fagus species. AT this point there are dozens of chloroplast genomes and it is well known that they are well conserved in most of the plant species studied. The novelty is inexistent.

Reviewer 2 ·

Basic reporting

This is a very good study concerning the assembled chloroplast genome of Fagus crenata. The manuscript is generally very clear and well-written although there are minor text revisions that I think the authors can still perform. For instance, the authors repeat across the text that the “assembled genome was a total of XXX bp” I see this sentence published in many studies but I still think the correct sentence would be “has a total of XXX bp”. Other minor typos should also be checked (e.g, line 99” following by manual modification”).

Figures and tables are clear and very well explained. Congratulations to the authors for this!

The main limitation of this paper is that the Discussion is very poor. In fact, there is almost no discussion. This is really a disappointment because the authors have clear, solid results that could be used to construct a very good Discussion. References are very scarce and limited.

Experimental design

The experimental design is overall correct but more details are needed. For instance, why was a NJ tree constructed? This seems to be a poor analysis in comparison to other widely-used methods (e.g, maximum likelihood in RAxML, Bayesian analysis in MrBayes). I also assumed that you only used the coding genes to built the tree….is that correct? Please clarify.

Where is the list of microsatellite regions? This should be part of the main text. There are no other SSRs above tri-nucleotide repeats?

Validity of the findings

Data is robust and sound but there is almost no Discussion. No clear conclusions are given.

Additional comments

This is a very good paper that can potentially be published in PeerJ. It has no flaws and it is overall very clear. The only weakness is the lack of a solid discussion.

Reviewer 3 ·

Basic reporting

The authors report on the sequencing, assembly, and annotation of the chloroplast genome of Fagus crenata (subgenus Fagus), and its comparison with the only other chloroplast genome available for the Fagus genus (F. engleriana subgenus Engleriana).

This is a very concise manuscript describing in detail the organization of the chloroplast. It is well written in a clear manner and professional English language is used throughout.

To the best of my knowledge, the Introduction and background cover the relevant literature.

There are some oversights on part of the authors regarding the availability of their data that hinder the full reproducibility of their work.

1. The authors should deposit the original short reads from their NGS experiment on an appropriate database (for example the SRA at NCBI).

2. The Multiple Sequence Alignments (MSA) of the full chloroplasts mentioned in line 102 and used to produce the phylogenetic tree shown on figure 2 should be included as supplementary data as a plain-text file in an appropriate format (FASTA aligned, Clustal, nexus, phylip, for example).

Experimental design

The methodology is, in general, of sufficient clarity and detail. There is only one point of their methodology that I consider should be carried out with more care: the phylogenetic inference presented on figure 2.
Firstly, the authors should be more precise when detailing the construction of the Multiple Sequence Alignment. Currently, it is only stated that MAFFT v7.308 was used (Line 102). From the previous line I assume that this was done inside Geneious v9.0.5 but no other information of the parameters used is stated. MAFFT v7.308 offers a range of multiple alignment methods (FFT-NS-i, E-INS-i, L-INS-i, G-INS-i and some others), each being a collection of parameters best suited for different situations. For the case of closely related chloroplasts either L-INS-i or G-INS-i could be the best option (this should be tested on their data and explicitly stated on the methodology section).
Secondly, the default parameters of the Neighbor Joining method under Geneious (or any other phylogenetic reconstruction software) should be adjusted for the Multiple Sequence Alignment used. The most important parameter being the DNA substitution model selected and its modifications (gamma parameter and/or invariable sites). These should be calculated through an appropriate estimator application (for example jModeltest2, prottest, or the one included in MEGA). It is probable that the structure or topology of their tree may not change but the resulting phylogeny will have a better estimate of individual branch lengths and total length of the tree (which also should be included on the description of the tree).
Lastly, although the tree on figure 2 shows bootstrap confidence values, no mention of how many repetitions were used to calculate those values. The information of how many total/informative positions were on the Multiple Sequence Alignment should also be included.

For the sake of helping visualizing the conservation/divergence of the chloroplast data I would suggest the following changes be made:

On lines 121-122 the authors state: “The two Fagus chloroplast genomes were relatively conserved (Figure 3) with the IR region more conserved than the LSC and SSC regions.” The IR, LSC, and SSC regions should be marked on it. Using Morella rubra as the reference makes it difficult to assess the relative conservation between both Fagus species. I would suggest making Fagus crenata the reference for this figure.

These same data could be summarized on a table showing the percent identities in a pair-wise manner (F. crenata vs F. enlgeriana, F. crenata vs M. rubra, and F. enlgeriana vs M. rubra) for the whole chloroplasts and for each of the regions. Also, the total gaps on the alignments should be included.

Optional minor detail:

The “most variable regions” (Lines 127-128) could be ordered from highest to lowest.

Validity of the findings

Although this reviewer understands the importance of reporting the assembly and annotation of the chloroplast genome for future studies into the inter- and intra-specific genetic structure and diversity of the genus Fagus, I encourage the authors to expand on its discussions about the implications of the divergence found on coding and non-coding regions, and the relevance of the SNPs and SSRs as possible markers.

Additional comments

No comment

---

## Round 0.2 · accepted · Accept

Dear authors

The reviewer has confirmed that you have addressed all the minor corrections and that the manuscript is now aceptable.
I congratulate you for the nice piece of work, which will add value to PeerJ.

# Reviewer 3 ·

Basic reporting

The authors have addressed all of my stated concerns.

Experimental design

The authors have addressed all of my stated concerns.

Validity of the findings

The authors have addressed all of my stated concerns.